# SPARE-Tau: A flortaucipir machine-learning derived early predictor of cognitive decline

Jon B. Toledo[1,2]*, Tanweer Rashid[3], Hangfan Liu[3,4], Lenore Launer[5], Leslie M. Shaw[6], Susan R. Heckbert[7], Michael Weiner[8,9,10,11,12], Sudha Seshadri[13], Mohamad Habes[3,4,13]*, for the Alzheimer's Disease Neuroimaging Initiative[¶]

1 Department of Neurology, University of Florida College of Medicine, Gainesville, Florida, United States of America, 2 Department of Neurology Houston Methodist Hospital, Houston, Texas, United States of America, 3 Neuroimage Analytics Laboratory (NAL) and the Biggs Institute Neuroimaging Core (BINC), Glenn Biggs Institute for Alzheimer's & Neurodegenerative Diseases, University of Texas Health Science Center San Antonio (UTHSCSA), San Antonio, Texas, United States of America, 4 Center for Biomedical Image Computing and Analytics (CBICA), University of Pennsylvania, Philadelphia, Pennsylvania, United States of America, 5 Laboratory of Epidemiology and Population Sciences, Intramural Research Program, National Institute on Aging, Bethesda, Maryland, United States of America, 6 Department of Pathology and Laboratory Medicine, Perelman School of Medicine, University of Pennsylvania, Philadelphia, Pennsylvania, United States of America, 7 Department of Epidemiology and Cardiovascular Health Research Unit, University of Washington, Seattle, Washington, United States of America, 8 Department of Veterans Affairs Medical Center, Center for Imaging of Neurodegenerative Diseases, San Francisco, California, United States of America, 9 Department of Radiology, University of California, San Francisco, California, United States of America, 10 Department of Medicine, University of California, San Francisco, California, United States of America, 11 Department of Psychiatry, University of California, San Francisco, California, United States of America, 12 Department of Neurology, University of California, San Francisco, California, United States of America, 13 Glenn Biggs Institute for Alzheimer's and Neurodegenerative Diseases, University of Texas Health Sciences Center, San Antonio, Texas, United States of America

¶ Membership of the Alzheimer's Disease Neuroimaging Initiative Group is listed in the Acknowledgments.
* habes@uthscsa.edu (MH); jtoledo@houstonmethodist.org (JBT)

**Data Availability Statement:** Data used in the preparation of this article were obtained from the Alzheimer's Disease Neuroimaging Initiative (ADNI) database (adni.loni.usc.edu). As such, the investigators within the ADNI contributed to the

## Abstract

### Background

Recently, tau PET tracers have shown strong associations with clinical outcomes in individuals with cognitive impairment and cognitively unremarkable elderly individuals. flortaucipir PET scans to measure tau deposition in multiple brain areas as the disease progresses. This information needs to be summarized to evaluate disease severity and predict disease progression. We, therefore, sought to develop a machine learning-derived index, SPARE-Tau, which successfully detects pathology in the earliest disease stages and accurately predicts progression compared to a priori-based region of interest approaches (ROI).

### Methods

587 participants of the Alzheimer's Disease Neuroimaging Initiative (ADNI) cohort had flortaucipir scans, structural MRI scans, and an Aβ biomarker test (CSF or florbetapir PET) performed on the same visit. We derived the SPARE-Tau index in a subset of 367 participants. We evaluated associations with clinical measures for CSF p-tau, SPARE-MRI, and flortaucipir PET indices (SPARE-Tau, meta-temporal, and average Braak ROIs). Bootstrapped multivariate adaptive regression splines linear regression analyzed the association between the biomarkers and baseline ADAS-Cog13 scores. Bootstrapped multivariate linear regression

design and implementation of ADNI and/or provided data but did not participate in the analysis or writing of this report. A complete listing of ADNI investigators can be found at: http://adni.loni.usc.edu/wp-content/uploads/how_to_apply/ADNI_Acknowledgement_List.pdf. Written consent was obtained from all ADNI participants and local IRBs at each site approved the study. For data access requests, the authors acknowledge the use of the ADNI cohort as suggested here, under point number 12: https://adni.loni.usc.edu/wp-content/uploads/how_to_apply/ADNI_Data_Use_Agreement.pdf.

**Funding:** This study was partly supported by the National Institutes of Health (NIH) grant numbers 1R01AG080821, P30AG066546, 1U24AG074855, and the San Antonio Medical Foundation grant SAMF – 1000003860. JBT is the Harrison Endowed Research Director at the Nantz National Alzheimer Center and has received support from the Edmond J. Safra Fellowship in Movement Disorders. Data collection and sharing for this project were funded by the Alzheimer's Disease Neuroimaging Initiative (ADNI) (National Institutes of Health Grant U01 AG024904) and DOD ADNI (Department of Defense award number W81XWH-12-2-0012). ADNI is funded by the National Institute on Aging, the National Institute of Biomedical Imaging and Bioengineering, and through generous contributions from the following: AbbVie, Alzheimer's Association; Alzheimer's Drug Discovery Foundation; Araclon Biotech; BioClinica, Inc.; Biogen; Bristol-Myers Squibb Company; CereSpir, Inc.; Cogstate; Eisai Inc.; Elan Pharmaceuticals, Inc.; Eli Lilly and Company; EuroImmun; F. Hoffmann-La Roche Ltd and its affiliated company Genentech, Inc.; Fujirebio; GE Healthcare; IXICO Ltd.; Janssen Alzheimer Immunotherapy Research & Development, LLC.; Johnson & Johnson Pharmaceutical Research & Development LLC.; Lumosity; Lundbeck; Merck & Co., Inc.; Meso Scale Diagnostics, LLC.; NeuroRx Research; Neurotrack Technologies; Novartis Pharmaceuticals Corporation; Pfizer Inc.; Piramal Imaging; Servier; Takeda Pharmaceutical Company; and Transition Therapeutics. The Canadian Institutes of Health Research is providing funds to support ADNI clinical sites in Canada. Private sector contributions are facilitated by the Foundation for the National Institutes of Health (www.fnih.org). The grantee organization is the Northern California Institute for Research and Education, and the study is coordinated by the Alzheimer's Therapeutic Research Institute at the University of Southern California. The funders had no role in study design, data collection and

models evaluated associations with clinical diagnosis. Cox-hazards and mixed-effects models investigated clinical progression and longitudinal ADAS-Cog13 changes. The Aβ positive cognitively unremarkable participants, not included in the SPARE-Tau training, served as an independent validation group.

## Results

Compared to CSF p-tau, meta-temporal, and averaged Braak tau PET ROIs, SPARE-Tau showed the strongest association with baseline ADAS-cog13 scores and diagnosis. SPARE-Tau also presented the strongest association with clinical progression in cognitively unremarkable participants and longitudinal ADAS-Cog13 changes. Results were confirmed in the Aβ+ cognitively unremarkable hold-out sample participants. CSF p-tau showed the weakest cross-sectional associations and longitudinal prediction.

## Discussion

Flortaucipir indices showed the strongest clinical association among the studied biomarkers (flortaucipir, florbetapir, structural MRI, and CSF p-tau) and were predictive in the preclinical disease stages. Among the flortaucipir indices, the machine-learning derived SPARE-Tau index was the most sensitive clinical progression biomarker. The combination of different biomarker modalities better predicted cognitive performance.

## Introduction

Alzheimer's disease (AD) is neuropathologically defined by the presence of tau neurofibrillary tangles and Aβ plaques [1]. Among these defining histopathological lesions, neurofibrillary tangles have been associated with a faster clinical progression than Aβ plaques [2, 3]. Tau has been historically measured on cerebrospinal fluid (CSF); however, this method does not provide sufficient information on the spatial distribution of tangle accumulation throughout the brain. On the other hand, Positron Emission Tomography (PET) advances offered several tau tracers, which have recently become available to quantify precise regional brain neurofibrillary tangle deposition. These new tracers can detect protein deposits present years before cognitive decline manifests. Tau tangles have been shown to capture stages of Alzheimer's disease [4], leading to diagnostic frameworks enabling the categorization of subjects along the AD continuum [5] using a biomarker-based definition of AD [5, 6].

Neuroimaging techniques capture changes across the whole brain that can be successfully summarized using machine-learning derived approaches [7–9]. Machine-learning algorithms generate optimal weighting for the different brain regions deriving summary indices with better classification accuracy and conversion predictions than simple anatomical-based summary metrics [10, 11]. Previous work has previously developed neuroimaging-based machine learning indices using magnetic resonance imaging (MRI) [7–9]. These indices have multiple uses in clinical practice and trials, in which they can facilitate recruitment and evaluate outcomes [12–14].

However, studies relied on a priori defined anatomical composites (i.e., meta-temporal regions of interest (ROI)), to evaluate the association with longitudinal outcomes [15–20]. This selection might not provide the optimal weighting of the individual brain regions involved throughout the disease. There is also limited information regarding biomarker-

analysis, decision to publish, or preparation of the manuscript.

**Competing interests:** Dr. Shaw provides quality control oversight for the Roche Electrosys immunoassay platform as part of the ADNI-3 study. Dr. Weiner has served on the Scientific Advisory Boards for Alzheon, Inc., Accera, Merck, Nestle (Nolan), PCORI (PPRN), Eli Lilly, Delfino Logic Ltd. (for Merck), Dolby Ventures, Brain Health Registry, and ADNI. He served on the editorial boards for Alzheimer's & Dementia and MRI. He has provided consulting and/or acted as a speaker/lecturer to Synarc, Pfizer, Accera, Inc., Alzheimer's Drug Discovery Foundation (ADDF), Merck, BioClinica, Eli Lilly, Howard University, Guidepoint, Denali Therapeutics, Nestle/Nestec, GLG Research, Atheneum Partners, BIONEST Partners, American Academy of Neurology (AAN), and Society for Nuclear Medicine and Molecular Imaging (SNMMI). This does not alter our adherence to PLOS ONE policies on sharing data and materials. Other authors report no competing interests.

associated outcomes [15]. In this work, we developed a new machine-learning derived tau PET index, the SPARE-Tau (Spatial Pattern of Abnormality for Recognition of Early Tau pathology) and compared it to previously established biomarkers. We evaluated the clinical associations and prognostic value of CSF p- tau, *a priori*-defined regional tau PET indices (meta-temporal ROI and average Braak score), and a machine learning-derived MRI index (SPARE-AD) [8, 9]. We hypothesize that [1] machine learning derived flortaucipir PET imaging composites offer stronger associations with cross-sectional and longitudinal clinical measures than a priori-defined tau PET ROIs, and [2] correlate better with clinical outcomes compared to MRI-defined indices and CSF p-tau.

## Methods

### Participants and clinical testing

Data used to prepare this article were obtained from the Alzheimer's Disease Neuroimaging Initiative (ADNI) database (adni.loni.usc.edu). The ADNI was launched in 2003 as a public-private partnership led by Principal Investigator Michael W. Weiner, MD. The primary goal of ADNI has been to test whether serial MRI, PET, other biological markers, and clinical and neuropsychological assessment can be combined to measure the progression of MCI and early AD. 587 ADNI participants with flortaucipir PET scans, MRI scans, and cerebrospinal fluid (CSF) A$\beta_{1-42}$ or florbetapir PET A$\beta$ testing during the same study visit were included (S1 Table). Our study included 344 cognitively unremarkable (CU), 182 MCI, and 61 dementia participants. Participants had yearly neuropsychological battery testing and clinical assessments [21]. The median follow-up was 1.9 years (IQR: 0.79–2.21 years). Further details on the clinical core, recruitment, and diagnostic methods have been previously published [22, 23], and details can be found at (http://adni.loni.usc.edu/). All the data is available at http://adni.loni.usc.edu/. Participants were stratified as normal (A$\beta$-) and pathological (A$\beta$+) A$\beta$ biomarker values if either their cerebrospinal fluid (CSF) or florbetapir PET scan indicated pathological A$\beta$ values (see PET and CSF sections below). The demographic and biomarker information of the participants I summarized in S1 Table. We downloaded the anonymized data from the ADNI website. Patient gave written informed consent; no minors were recruited into the study. The study was approved by the local institutional review boards (IRBs).

### MRI acquisition and processing

3T sagittal MP-RAGE scans for each subject were selected at the same clinical visit as the flortaucipir scan and were segmented and parcellated with Freesurfer (v 5.3) [24]. Additional details for the imaging processing can be found on the ADNI website (http://www.adni-info.org/). The Spatial Pattern of Abnormality for Recognition of Early Alzheimer's disease [25] (SPARE-AD) index is a previously validated imaging signature used to estimate Alzheimer's disease-like atrophy patterns in the brain [8, 11]. A support vector machine (SVM) was used to differentiate between dementia and CU participants maximally. The SVM classifier with a linear kernel was trained with structural MR scans to classify participants as dementia and CU. The training data included only healthy controls with known-negative A$\beta$ status and only dementia participants with known-positive A$\beta$ status. Higher positive SPARE-AD values indicate a more Alzheimer's disease-like brain structure, and lower negative values indicate normal brain structure. The SPARE-BA model was trained with CU data only and applied to all participants included in this study. A model having a radial basis function kernel was evaluated with leave-one-out cross-validation using structural region of interest volumes from 352 CU participants and had a mean absolute error of 4.22. The predicted brain age for the CU participants was then adjusted for age using a linear regression model, like previous work [9].

## PET acquisition and processing

For the flortaucipir PET scans, 370 MBq (10.0 mCi) ± 10% of 18F-flortaucipir were administered, with 30-minute (6X5 minutes frames) acquisition at 75–105 min post-injection. Each flortaucipir scan was co-registered to its corresponding MP-RAGE scan, and mean flortaucipir uptake within each Freesurfer-defined brain region was calculated. Data were corrected for partial volume effects using the geometric transfer matrix approach1. Mean regional uptake was normalized by inferior cerebellar gray matter as a reference region to generate the flortaucipir SUVRs. Further information can be found on the ADNI website (http://adni.loni.usc.edu/). We included partial volume corrected ROIs, which were normalized to the inferior cerebellum. Meta-temporal ROI was calculated as previously described (see supplementary material). The average Braak score was calculated as the average of Braak I, Braak III-IV, and Braak V-VI areas.

For the florbetapir PET scans, 370 MBq (10.0 mCi) ± 10% of 18F-florbetapir were administered, with 20 minutes (4X minute frames) acquisition at 50–70 min post-injection. SPM8 software was used to co-register the florbetapir PET scans with the corresponding MRI scans. Florbetapir means from the gray matter in subregions were extracted within four large regions (frontal, anterior/posterior cingulate, lateral parietal, lateral temporal) [26, 27], and weighted means for each of the four main regions were created. A composite was used as a reference region, based on the whole cerebellum, brainstem/pons, and eroded subcortical white matter (http://www.adni-info.org/). A value ≥0.78 in the summary composite florbetapir index classified participants as Aβ+.

**SPARE-AD training.** A support vector machine (SVM) was used to differentiate between dementia and CU participants maximally. The SVM classifier with a linear kernel was trained with structural MR scans to classify participants as dementia and CU. The training data included only healthy controls with known-negative Aβ status and only dementia participants with known-positive Aβ status. Higher positive SPARE-AD values indicate a more Alzheimer's disease-like brain structure, and lower negative values indicate normal brain structure.

**The SPARE-Tau index.** A classification model using a support vector machine (SVM) with a linear kernel was developed and trained to predict the clinical status of 367 participants defined as control group (n = 218, CU individuals with normal Aβ biomarker values) or pathologic group (n = 149, MCI and dementia individuals with pathological Aβ values). The model was trained with 50-fold cross-validation and used the Freesurfer parcellated ROIs' SUVR values. Similar machine learning models have been previously described and validated on MRI [8, 11]. More positive SPARE-Tau indices indicate pathological tau deposition, and more negative indices imply lower tau deposition. Areas included in the final model are summarized in S1 Fig.

## Cerebrospinal fluid collection and Aβ1−42 measurements

CSF samples were obtained in the morning after an overnight fast and processed as previously described [28, 29] (http://adni.loni.usc.edu/). Roche Elecsys $Aβ_{1−42}$ and tau CSF immunoassay measurements were performed at the UPenn/ADNI biomarker laboratory following the Roche Study protocol [22]. The cutoff for pathological values was 977 pg/mL for $Aβ_{1−42}$ and 27 pg/mL for p-tau [30]. Measurements performed during the same ADNI visit as the flortaucipir scans were selected (12 days median time interval between CSF draw and PET scans).

## Statistical analysis

We calculated median and interquartile range (IQR) values to summarize quantitative variables and proportions for categorical variables. Kruskal-Wallis analyses and chi-square tests

were applied to compare continuous and categorical variables between the groups. Spearman rank correlations evaluated the associations between the different measures. CU participants with normal Aβ biomarker values and MCI and dementia individuals with pathological Aβ values were included in the SPARE-Tau training. CU participants with pathological Aβ biomarker values were not used in the training of the SPARE-Tau index and therefore served as an independent testing group. Multivariate analyses included standardized biomarker values to compare the coefficients. We applied multivariate adaptive regression splines (MARS) models to evaluate the association between the different biomarker values. For each biomarker, we performed 1,000 bootstraps with replacement. We analyzed 1,000 bootstrapped linear regression models with biomarker values as dependent variables and age, gender, education, and clinical diagnosis as predictors. We compared the $R^2$ and coefficients values from the bootstrapped models using Friedman tests, followed by post-hoc comparisons with Wilcoxon signed-rank tests to evaluate which biomarker offered the best fit. A linear discriminant model with 10-fold cross-validation identified cutoffs to define normal and pathological tau PET indices and SPARE-AD scores used in longitudinal analyses. Cox hazards models evaluated the progression from CU to MCI (sex, age, and education included as covariates). We used mixed-effects models that included ADAS-Cog13 as the outcome to evaluate longitudinal disease progression. These models included time, sex, age, education, clinical diagnosis, and biomarkers as fixed effects. We included clinical diagnosis and biomarkers interactions with time. Participants and time were included as random effects. Power transformations were used in parametric analyses as needed to achieve normal distribution. P-values <0.05 (two-sided) were considered statistically significant. Bonferroni-Holm multiple comparison correction was applied to correct for multiple comparisons and the post hoc comparisons. Analyses were performed using R version 4.2.

## Results

### Correlation between AD biomarkers

We evaluated correlations between biomarkers included in this study (SPARE-Tau, average Braak areas, meta-temporal ROI, CSF p-tau, SPARE-AD, and florbetapir composite score) in groups stratified by Aβ status. Associations were stronger in the Aβ+ participants than in the Aβ- participants (Fig 1A and 1B). Aβ+ participants showed strong correlations between tau PET indices and moderate correlations of the tau PET indices with the other biomarkers (CSF p-tau, SPARE-AD, and florbetapir composite score). Aβ- participants presented moderate correlations between the different tau indices, but correlations with the other biomarkers (CSF p-tau, SPARE-AD, and florbetapir composite score) were weak or absent (≤0.25).

### Baseline clinical associations

SPARE-Tau best explained ADAS-Cog13 values in the Aβ+ participants when we compared the $R^2$ values (explained ADAS-Cog13 variance) of the bootstrapped MARS splines (Fig 1C and 1D, S2 and S3 Tables). In the Aβ- participants, only the florbetapir summary composite showed a similar association with ADAS-Cog-13 (Fig 1D and S3 Table) as SPARE-Tau. In contrast, all other indices explained lower ADAS-Cog13 variance (p-value<0.0001). Combining SPARE-Tau and SPARE-AD (global 1) led to an increase in the explained ADAS-Cog13 variance in the Aβ+ ($R^2$ difference 0.14, p-value<0.00001) and Aβ- participants ($R^2$ difference 0.10, p-value<0.0001). Further adding the florbetapir summary composite (global 2) led to an increase in the explained ADAS-Cog13 variance in the Aβ- participants ($R^2$ difference 0.14, p-value<0.00001), with a minimal but significant improvement in the Aβ+ participants ($R^2$

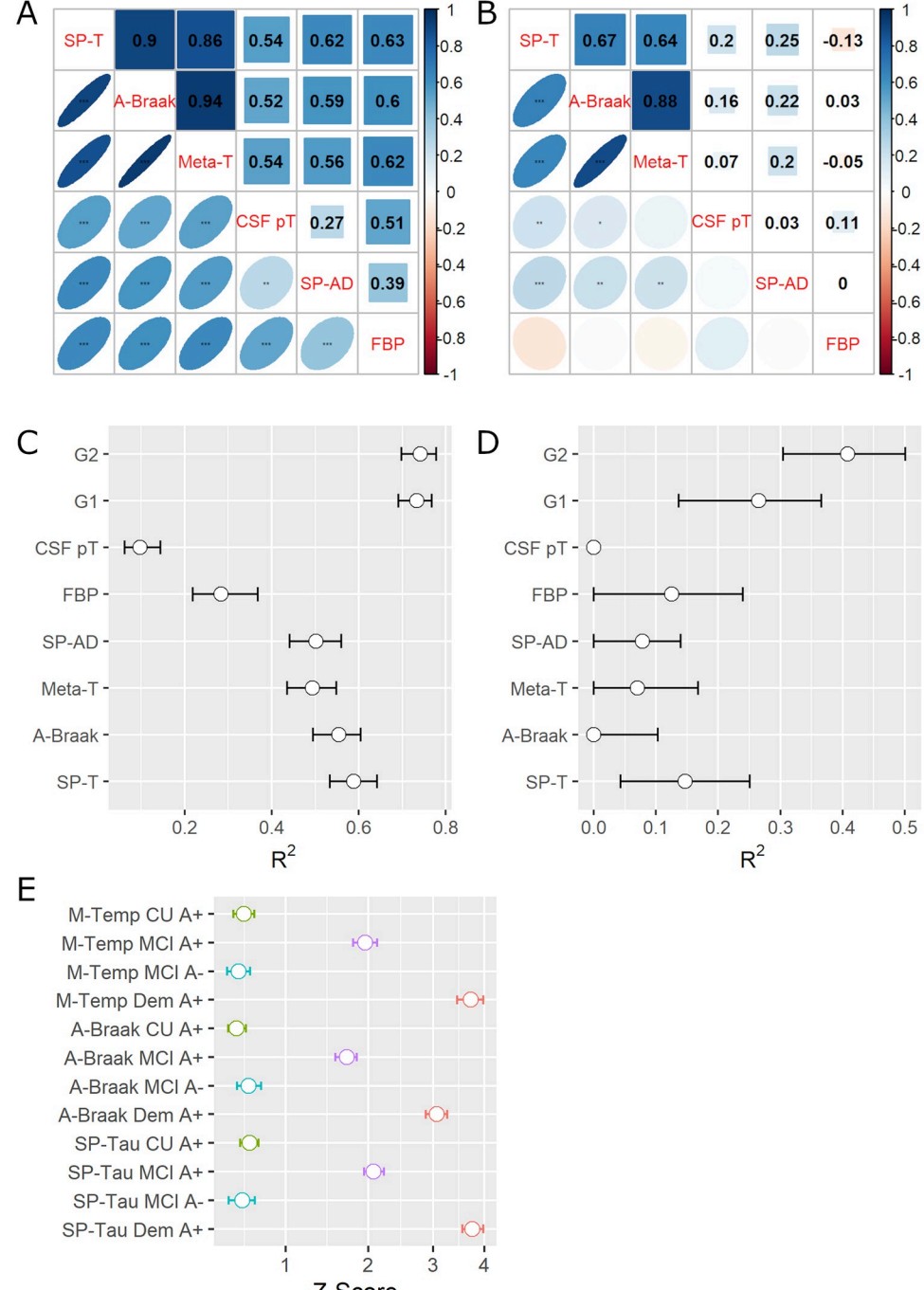

**Fig 1. Evaluation of biomarker indices (SPARE-Tau, Average Braak, Meta-Regions, CSF p-tau, SPARE-AD and florbetapir composite score).** (A) Correlations between the different biomarkers in the Aβ+ participants (*: p-value<0.05; *: p-value<0.01; *: p-value<0.0001). (B) Correlations between the different biomarkers in the Aβ-participants (*: p-value<0.05; *: p-value<0.01; *: p-value<0.0001). (C) Coefficient of determination ($R^2$) of MARS spline models predicting baseline ADAS-Cog13 values in all the Aβ+ participants. Global 1 (G1) includes SPARE-Tau and SPARE-AD as predictors. Global 2 (G2) combines SPARE-Tau, SPARE-AD, and the florbetapir composite score (FBP). (D) Coefficient of determination ($R^2$) of MARS spline models predicting baseline ADAS-Cog13 values in all the Aβ- participants. Global 1 (G1) includes SPARE-Tau and SPARE-AD as predictors. Global 2 (G2) combines SPARE-Tau, SPARE-AD, and the florbetapir composite score (FBP). (E): Coefficients of z-scored flortaucipir indices comparing the clinical groups stratified by Aβ biomarker values against Aβ- cognitively unremarkable (CU) participants. A-Braak: Average Braak-area flortaucipir score; CSF pT: CSF p-tau; Dem: Dementia; FBP: florbetapir composite; MCI: Mild cognitive impairment; Meta-T: Meta-temporal ROI flortaucipir score; SP-AD: SPARE-AD; SP-T: SPARE-Tau.

**Table 1. Prediction of clinical progression in CU participants.** Analyses adjusted by age, education, and sex. Biomarkers values were classified as normal or pathological to predict clinical progression. Bonferroni-Holm multiple comparison correction applied.

| | CU (All) to MCI/Dementia ($n_{Total}$ = 331, $n_{Progressors}$ = 20) | | CU (Aβ+) to MCI/Dementia ($n_{Total}$ = 122, $n_{Progressors}$ = 13) | |
|---|---|---|---|---|
| | H.R. (95% CI) | p-value | H.R. (95% CI) | p-value |
| Flortaucipir SPARE-Tau index | 7.9 (2.7–22.7) | 0.0006 | 6.2 (1.7–22.8) | 0.022 |
| Flortaucipir meta-temporal ROI index | 3.3 (1.2–8.8) | 0.021 | 2.79 (0.87–9.0) | 0.11 |
| Flortaucipir average Braak index | 5.3 (1.9–14.7) | 0.003 | 4.6 (1.4–15.6) | 0.022 |
| Florbetapir Composite index | 3.1 (0.97–9.7) | 0.056 | - | - |
| MRI SPARE-AD | 3.9 (1.5–10.1) | 0.009 | 4.9 (1.4–17.1) | 0.022 |

H.R.: Hazard ratio.

difference 0.009, p-value<0.0001). We excluded CSF p-tau from further analyses due to its weak association with the clinical measures.

All flortaucipir indices were higher in Aβ+ participants (including the Aβ+ CU group for the SPARE-Tau and meta-temporal ROI), with a progressive increase in the Aβ+ MCI and dementia participants (Fig 1E). SPARE-Tau presented the highest z-scored differences in all the Aβ+ groups compared to the Aβ- CU group (p-value<0.0001). The average Braak score showed the highest value for the Aβ- MCI group (p<0.0001) and also was the only index that showed higher values in the Aβ- MCI than the Aβ+ CU group. SPARE-Tau showed the highest $R^2$ (0.48, IQR = 0.45–0.51), compared to average Braak ($R^2$ = 0.41, IQR = 0.38–0.44) and meta-temporal ROI ($R^2$ = 0.41, IQR = 0.38–0.44).

## Longitudinal clinical associations

To evaluate the association with the longitudinal changes, we estimated SPARE-Tau, average Braak score, meta-temporal ROI, and SPARE-AD cutoffs based on classifying CU Aβ- participants versus Aβ+ MCI and dementia participants. For the florbetapir Aβ PET, we used the previously derived florbetapir composite score.

All the biomarkers predicted progression from CU to MCI/dementia when all the CU participants were included (Table 1), but when we evaluated the clinical progression in the Aβ + CU participants, the meta-temporal ROI did not predict clinical progression, and SPARE-Tau remained the strongest association. All three flortaucipir PET measures and SPARE-AD predicted longitudinal changes in ADAS-Cog13 in the whole cohort (Table 2), but only SPARE-Tau predicted longitudinal changes in the Aβ+ CU participants. None of the biomarkers

**Table 2. Prediction of ADAS-Cog13 longitudinal changes.** Analyses adjusted by age, education, and sex. Model with all participants also adjusted for baseline clinical diagnosis. Biomarkers values were classified as normal or pathological to predict cognitive decline. Bonferroni-Holm multiple comparison correction applied.

| | All CU Participants (n = 172) | | | CU Aβ- (n = 100) | | | CU Aβ+ (n = 72) | | |
|---|---|---|---|---|---|---|---|---|---|
| | Coef. (SE) | p-value | AIC | Coef. (SE) | p-value | AIC | Coef. (SE) | p-value | AIC |
| Flortaucipir SPARE-Tau index | 0.35 (0.05) | <0.0001 | 3074.3 | 0.16 (0.15) | 0.32 | 980.7 | 0.28 (0.09) | 0.0070 | 713.8 |
| Flortaucipir meta-temporal ROI index | 0.32 (0.05) | <0.0001 | 3094.8 | 0.16 (0.12) | 0.24 | 980.9 | 0.20 (0.1) | 0.071 | 716.9 |
| Flortaucipir average Braak index | 0.26 (0.05) | <0.0001 | 3103.7 | 0.14 (0.12) | 0.31 | 981.5 | 0.18 (0.1) | 0.12 | 716 |
| MRI SPARE-AD | 0.19 (0.05) | 0.0002 | 3129.2 | 0.14 (0.09) | 0.20 | 978.9 | 0.10 (0.11) | 0.36 | 728.7 |

AIC: Akaike information criterion. Coef.: Coefficient; SE: Standard error.

predicted longitudinal changes in the Aβ- CU participants. A comparison of SPARE-Tau and CSF p-tau is included in S4 Table.

## Discussion

Among the three tested flortaucipir measures (SPARE-Tau, meta-temporal ROI, and average Braak score), our novel SPARE-Tau index offered the best classification accuracy. SPARE-Tau showed the largest differences between the Aβ+ and the Aβ- CU participants, best-predicted baseline ADAS-Cog13 scores, and presented the strongest association with longitudinal clinical progression (including the CU Aβ+ participants).

AD biomarker models and studies of participants with AD autosomal dominant mutations indicate that Aβ biomarkers precede tau biomarkers [4, 31]. About 30% of CU elderly individuals are Aβ+ in the seventh decade of life [32, 33]. In turn, tau changes are closer to the onset of cognitive decline and have been considered a marker for the disease [5]. Neuropathological studies showed a stronger association of tau pathology with cognition and explained a large part of cognitive changes present in cognitively impaired individuals compared to other individuals [34]. Flortaucipir binding correlates with neurofibrillary tangle deposition in AD and regional neurofibrillary pathology burden [35]. Therefore, we expected tau PET tracers to outperform Aβ biomarkers to predict clinical outcomes. Imaging-based biomarkers reflect changes across the whole brain. This information needs to be summarized to facilitate its clinical application. Previous flortaucipir PET measures have been developed on averages of ROI [27, 36]. This follows previous MRI approaches that identified hippocampal atrophy as a measure of neurodegeneration in AD. A limitation of these analyses is that they select a subset of the regions and do not weigh them according to their importance. We previously developed a support vector machine-derived MRI index, SPARE-AD, which showed improved classification and prediction of clinical progression compared to ROI-based MRI indices [10]. Here we expanded the SPARE framework to include the SPARE-Tau index. These machine-learning approaches combine the information derived from multiple brain regions to provide a global, easily interpretable, sensitive and specific measures compared to single ROI, like the hippocampus.

Flortaucipir has shown an inverse correlation with brain atrophy, stronger than the one observed for Aβ PET scans [17, 18], in line with our finding. We identified a correlation (r = 0.62) between our SPARE-AD and SPARE-Tau indices. Our previously developed MRI index (SPARE-AD) underperformed all flortaucipir indices when evaluating clinical progression and cognitive decline. This finding might be counterintuitive because structural MRI reflects atrophy related to AD-specific regions, and those might be injured later in the AD timing model [37]. Additionally, potential interactions with cognition should be studied in future work, evaluating *in-vivo* the different mechanisms of tau-related cognitive impairment (local structural damage versus functional network dysfunction). Nevertheless, neuropathological studies indicate that AD pathology is the primary driver of cognitive impairment [38].

Among the flortaucipir indices, the meta-temporal ROI (or other ROIs) is the most commonly used measure when clinical associations of flortaucipir scans are evaluated [15, 19, 35, 39]. We also included an index reflecting global flortaucipir burden, the average Braak index, based on the staging defined by Braak [40]. We evaluated several cross-sectional metrics (clinical diagnostic accuracy and ADAS-Cog13) and clinical progression (clinical progression of CU and MCI participants and cognitive decline measured using ADAS-Cog13). SPARE-Tau outperformed these commonly used ROI-based indices (meta-temporal ROI and the average Braak indices).

Moreover, SPARE-Tau identified the largest effect size difference when we compared Aβ+ CU participants (not used for training) to Aβ- CU participants and was the strongest predictor of clinical progression and cognitive decline in the Aβ+ CU participants (hold-out validation group). We also stratified our analyses by Aβ status, analyzing Aβ- and Aβ+ separately in several analyses, whereas our training groups evaluated CU Aβ- versus cognitively impaired Aβ+ participants. CSF p-tau underperformed all flortaucipir indices in our cross-sectional analyses, and we, therefore, excluded it from the longitudinal analyses. It can be expected that CSF tau measurements underperform ligand-based PET tau estimates as CSF tau represents a more indirect measure of overall brain tau deposition, and tau is deposited intracellularly in the form of neurofibrillary tangles. Other studies found a stronger cross-sectional association of PET ROI metrics with clinical than those observed for CSF tau assays [41]. Alternatively, it is possible that CSF p-tau identifies changes at an earlier preclinical stage than SPARE-Tau (23.6% abnormal CSF p-tau and 12% abnormal SPARE-Tau in the CU Aβ+ group). This could also explain why CSF p-tau underperforms SPARE-Tau in the case of a short follow-up. One study has described inconsistent findings of CSF p-tau better predicting cognition in CU participants than tau PET [42]. These differences might be to differences in cohort composition, CSF assays, and length of follow-up. Further studies with longer longitudinal follow-up that include plasma, CSF and PET tau measures in CU participants in CU participants are needed.

We expand the previous findings by additionally evaluating with the ADAS-Cog13 scale, predicting clinical conversion in CU participants, and assessing CSF p-tau, which surprisingly showed the lowest clinical associations. One recent study evaluated the longitudinal correlates of structural MRI and flortaucipir PET [15]. This study indicated that meta-temporal flortaucipir ROI showed the strongest association with longitudinal MMSE scores, followed by MRI (using predefined temporal lobe ROI), and least associated with Aβ PETs. Adding MRI information led to increased MMSE variability explained by the biomarkers. The authors acknowledged several limitations, like the lack of more detailed clinical measures, the lack of diagnostic conversion outcomes, and the need to evaluate biofluid biomarkers. Other studies have considered flortaucipir scans in preclinical stages, selecting a single ROI and identifying longitudinal clinical decline based on increased uptake in a single ROI [16, 43]. In addition, we included sophisticated machine-learning derived measures that improve the diagnostic performance over *a priori*-defined ROIs. The meta-temporal ROI also underperformed the average Braak score. We also confirmed that including our MRI measure, SPARE-AD, improved the model evaluating longitudinal ADAS-Cog13 changes; however, when we looked at the model's different components, SPARE-AD only showed an association with baseline ADAS-Cog13 values and was not associated with longitudinal ADAS-Cog13 changes. In agreement with previous neuropathological studies and disease models, recent studies have confirmed that flortaucipir PET scans (based on predefined ROIs) have a stronger association with longitudinal outcomes than Aβ PET scans [16, 43].

The finding of larger SPARE-Tau and average tau PET changes during follow-up in Aβ + participants agrees with recent findings of a ceiling effect in lower Braak stage regions as disease progresses [44]. Therefore, it is expected that indices that track areas beyond the temporal lobe will identify AD-related tau deposition better.

This manuscript's strengths are the large sample size, the comparison of multiple tau indices (including CSF and PET), florbetapir composite, and MRI structural measures using machine-learning derived indices. We also evaluated ADAS-Cog13 which offers more information than MMSE, used in other analyses, and we evaluated longitudinal outcomes. There are several limitations to our study: first, only a small number of participants progressed from CU to MCI (Table 1). Second, CSF tau and florbetapir scans were not available for all participants. Finally, although we used leave one out cross-validation, a commonly used procedure to

ensure generalization of results, there was no independent validation cohort accessible to us to confirm our results. Furthermore, we designed our study to leave Aβ+ CU participants out of the training sample and this sample served as an independent sample test sample.

This manuscript presents a novel machine-learning derived flortaucipir index that outperforms other previously utilized flortaucipir indices in multiple cross-sectional and longitudinal clinical outcomes, detecting changes and better prognosticating changes in the preclinical disease stages. We further compared its performance to other biomarker modalities, confirming that SPARE-Tau showed the best prediction and that MRI, but not florbetapir, added value to predicting baseline cognitive scores in Aβ+ participants.

## Supporting information

**S1 Fig. Brain regions contributing to SPARE-Tau index.**
(DOCX)

**S1 Table. Clinical and biomarker characteristics stratified by clinical diagnosis at baseline in the training participant group (top: CU with normal Aβ biomarker values and cognitively impaired with pathological Aβ biomarker values) and the early preclinical validation group (CU with pathological Aβ biomarker values).**
(DOCX)

**S2 Table. Cross-sectional prediction of ADAS-Cog13 scores using multivariate adaptive regression splines models.** Table presents mean $R^2$ and 95% CI of the 1000 bootstrapped models.
(DOCX)

**S3 Table. Comparison of baseline fit of ADAS-Cog13 scores ($R^2$ values) for each pair of multivariate adaptive regression splines models.** Paired t-test analysis. Bonferroni-Holm multiple comparison correction.
(DOCX)

**S4 Table. Association of baseline CSF p-tau and SPARE-Tau with longitudinal ADAS-Cog13 changes in the participant subset with CSF samples.** The top part of the table represents prediction based on cutoffs, and the bottom part evaluates z-scored biomarker values.
(DOCX)

## Acknowledgments

A full list of authors can be found at: https://adni.loni.usc.edu/wp-content/uploads/how_to_apply/ADNI_Acknowledgement_List.pdf.

## Author Contributions

**Conceptualization:** Jon B. Toledo, Mohamad Habes.

**Formal analysis:** Jon B. Toledo, Tanweer Rashid.

**Investigation:** Jon B. Toledo.

**Methodology:** Tanweer Rashid, Hangfan Liu, Mohamad Habes.

**Resources:** Mohamad Habes.

**Software:** Tanweer Rashid, Hangfan Liu.

**Supervision:** Mohamad Habes.

**Writing – original draft:** Jon B. Toledo, Mohamad Habes.

**Writing – review & editing:** Jon B. Toledo, Tanweer Rashid, Hangfan Liu, Lenore Launer, Leslie M. Shaw, Susan R. Heckbert, Michael Weiner, Sudha Seshadri, Mohamad Habes.

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
