## [Decision Letter · Decision Letter 0]

20 Sep 2022

PONE-D-22-21443SPARE-Tau: A Flortaucipir Machine-Learning Derived Early Predictor of Cognitive DeclinePLOS ONE

Dear Dr. Habes,

Thank you for submitting your manuscript to PLOS ONE. After careful consideration, we feel that it has merit but does not fully meet PLOS ONE’s publication criteria as it currently stands. Therefore, we invite you to submit a revised version of the manuscript that addresses the points raised during the review process.

We look forward to receiving your revised manuscript.

Kind regards,

Kensaku Kasuga

Academic Editor

PLOS ONE

Journal Requirements:

"JBT is supported by the Edmond J. Safra Fellowship in Movement Disorders. Data collection and sharing for this project were funded by the Alzheimer's Disease Neuroimaging Initiative (ADNI) (National Institutes of Health Grant U01 AG024904) and DOD ADNI (Department of Defense award number W81XWH-12-2-0012). ADNI is funded by the National Institute on Aging, the National Institute of Biomedical Imaging and Bioengineering, and through generous contributions from the following: AbbVie, Alzheimer’s Association; Alzheimer’s Drug Discovery Foundation; Araclon Biotech; BioClinica, Inc.; Biogen; Bristol-Myers Squibb Company; CereSpir, Inc.; Cogstate; Eisai Inc.; Elan Pharmaceuticals, Inc.; Eli Lilly and Company; EuroImmun; F. Hoffmann-La Roche Ltd and its affiliated company Genentech, Inc.; Fujirebio; GE Healthcare; IXICO Ltd.; Janssen Alzheimer Immunotherapy Research & Development, LLC.; Johnson & Johnson Pharmaceutical Research & Development LLC.; Lumosity; Lundbeck; Merck & Co., Inc.; Meso Scale Diagnostics, LLC.; NeuroRx Research; Neurotrack Technologies; Novartis Pharmaceuticals Corporation; Pfizer Inc.; Piramal Imaging; Servier; Takeda Pharmaceutical Company; and Transition Therapeutics. The Canadian Institutes of Health Research is providing funds to support ADNI clinical sites in Canada. Private sector contributions are facilitated by the Foundation for the National Institutes of Health (www.fnih.org). The grantee organization is the Northern California Institute for Research and Education, and the study is coordinated by the Alzheimer’s Therapeutic Research Institute at the University of Southern California. ADNI data are disseminated by the Laboratory for Neuro Imaging at the University of Southern California."

"I have read the journal's policy and the authors of this manuscript have the following competing interests:

Dr. Shaw provides quality control oversight for the Roche Electrosys immunoassay platform as part of the ADNI-3 study. Dr. Weiner has served on the Scientific Advisory Boards for Alzheon, Inc., Accera, Merck, Nestle (Nolan), PCORI (PPRN), Eli Lilly, Delfino Logic Ltd. (for Merck), Dolby Ventures, Brain Health Registry, and ADNI. He served on the editorial boards for Alzheimer's & Dementia and MRI. He has provided consulting and/or acted as a speaker/lecturer to Synarc, Pfizer, Accera, Inc., Alzheimer's Drug Discovery Foundation (ADDF), Merck, BioClinica, Eli Lilly, Howard University, Guidepoint, Denali Therapeutics, Nestle/Nestec, GLG Research, Atheneum Partners, BIONEST Partners, American Academy of Neurology (AAN), and Society for Nuclear Medicine and Molecular Imaging (SNMMI). Other authors report no competing interests."

6. One of the noted authors is a group or consortium "the Alzheimer’s Disease Neuroimaging Initiative*". In addition to naming the author group, please list the individual authors and affiliations within this group in the acknowledgments section of your manuscript. Please also indicate clearly a lead author for this group along with a contact email address.

7. Please amend your list of authors on the manuscript to ensure that each author is linked to an affiliation. Authors’ affiliations should reflect the institution where the work was done (if authors moved subsequently, you can also list the new affiliation stating “current affiliation:….” as necessary).

8. Please include your full ethics statement in the ‘Methods’ section of your manuscript file. In your statement, please include the full name of the IRB or ethics committee who approved or waived your study, as well as whether or not you obtained informed written or verbal consent. If consent was waived for your study, please include this information in your statement as well. 

Additional Editor Comments:

Please respond to reviewer's comments.

Reviewers' comments:

Reviewer's Responses to Questions

**Comments to the Author**

1. Is the manuscript technically sound, and do the data support the conclusions?

Reviewer #1: Yes

Reviewer #2: Yes

2. Has the statistical analysis been performed appropriately and rigorously? 

Reviewer #1: Yes

Reviewer #2: Yes

3. Have the authors made all data underlying the findings in their manuscript fully available?

Reviewer #1: Yes

Reviewer #2: Yes

4. Is the manuscript presented in an intelligible fashion and written in standard English?

Reviewer #1: Yes

Reviewer #2: Yes

5. Review Comments to the Author

Reviewer #1: This is well conducted study.

The author showed, Flortaucipir indices showed the strongest clinical association among the studied biomarkers and were predictive in the preclinical disease stages.

Among the flortaucipir indices, the machine-learning derived SPARE-Tau index was the most sensitive clinical progression biomarker.

The combination of different biomarker modalities better predicted cognitive performance.

Reviewer #2: The authors developed the novel machine learning derived index, SPARE-Tau, to assess tau PET accumulation in the earliest disease stage of Alzheimer’s disease (AD) continuum. The authors demonstrated that SPARE-Tau was the most sensitive clinical progression biomarker among all tested tau PET indices. This paper proposed the novel unbiased index of tau PET with flortaucipir as predictive biomarker in the preclinical disease stages of AD continuum to facilitate clinical research and clinical trials. However, there remains several concerns that should be addressed before considering publication.

Major points.

1. Previous studies reported that CSF p-Tau best predicted longitudinal cognitive decline in cognitively unimpaired subjects [X], while the authors showed no significant association of baseline CSF p-tau with longitudinal cognitive decline. Although predictive performance of tau PET could become better using the novel machine learning-derived index (SPARE-Tau) instead of other conventional indices, the reason why the predictive performance of CSF p-Tau discrepancies between the present and previous studies should be discussed.

X. Mattsson-Carlgren N, et al., Neurology. 2020;94(21):e2233-e2244.

2. Recent study demonstrated that CSF p-tau generally becomes abnormal before tau PET and that cognitively unimpaired subjects with abnormal CSF p-Tau and positive PET scan have a risk of clinical progression, followed by cognitively unimpaired subjects with abnormal CSF p-Tau despite negative scan of Tau PET [Y]. In the present study, the median follow up was just 1.9 years, which is relatively short duration. Taken together, although we can conclude that elevated accumulation of Tau PET must predict future cognitive decline, we didn’t draw the robust conclusion whether the CSF p-Tau would predict future cognitive decline because of the short follow up period.

Y. Groot C, et al., Brain. 2022 Sep 9:awac329. doi: 10.1093/brain/awac329. Online ahead of print.

3. In the section of “MRI Acquisition and Processing” (page 7, lines 128-131), the authors explained that “The training data included only healthy controls with known-negative Aβ status and only dementia participants with known-positive Aβ status.”; however, then it is also written that “The model was trained with CU data only and applied to all participants included in this study”. What does it mean? Was SPARE-AD model retrained without data of dementia participants with known-positive Aβ status?

Minor points

4. Page 6, lines 115-117

Regarding the PET reading method, whether visual judgment was used or semi-quantitative values such as centiloid scale or SUVR were used, and if so, how the cutoff values were set should be briefly indicated.

5. Page 8, lines 149-151

The “)” may be missing in the sentence.

6. Page 8, line 155

Only the title of the chapter (SPARE-AD Training) is given, the contents is not written.

7. Page 10, line 201

“CSF p=tau” shold be corrected to “CSF p-tau“.

6. PLOS authors have the option to publish the peer review history of their article (what does this mean?). If published, this will include your full peer review and any attached files.

Reviewer #1: No

Reviewer #2: **Yes: **Hitoshi SHIMADA

---

## [Author Response · Author response to Decision Letter 0]

4 Oct 2022

We added the following statement in the methods sections to answer this request: 

“We downloaded the anonymized data from the ADNI website. Patient gave written informed consent; no minors were recruited into the study.”

We now only include the grant funding information in the funding section of the submission system and excluded it from the manuscript. We also deleted the conflicts of interest section from the manuscript. 

"JBT is supported by the Edmond J. Safra Fellowship in Movement Disorders. Data collection and sharing for this project were funded by the Alzheimer's Disease Neuroimaging Initiative (ADNI) (National Institutes of Health Grant U01 AG024904) and DOD ADNI (Department of Defense award number W81XWH-12-2-0012). ADNI is funded by the National Institute on Aging, the National Institute of Biomedical Imaging and Bioengineering, and through generous contributions from the following: AbbVie, Alzheimer’s Association; Alzheimer’s Drug Discovery Foundation; Araclon Biotech; BioClinica, Inc.; Biogen; Bristol-Myers Squibb Company; CereSpir, Inc.; Cogstate; Eisai Inc.; Elan Pharmaceuticals, Inc.; Eli Lilly and Company; EuroImmun; F. Hoffmann-La Roche Ltd and its affiliated company Genentech, Inc.; Fujirebio; GE Healthcare; IXICO Ltd.; Janssen Alzheimer Immunotherapy Research & Development, LLC.; Johnson & Johnson Pharmaceutical Research & Development LLC.; Lumosity; Lundbeck; Merck & Co., Inc.; Meso Scale Diagnostics, LLC.; NeuroRx Research; Neurotrack Technologies; Novartis Pharmaceuticals Corporation; Pfizer Inc.; Piramal Imaging; Servier; Takeda Pharmaceutical Company; and Transition Therapeutics. The Canadian Institutes of Health Research is providing funds to support ADNI clinical sites in Canada. Private sector contributions are facilitated by the Foundation for the National Institutes of Health (www.fnih.org). The grantee organization is the Northern California Institute for Research and Education, and the study is coordinated by the Alzheimer’s Therapeutic Research Institute at the University of Southern California. ADNI data are disseminated by the Laboratory for Neuro Imaging at the University of Southern California."

We excluded all funding sources from the acknowledgements. We added the requested statement in the cover letter. 

"I have read the journal's policy and the authors of this manuscript have the following competing interests:

Dr. Shaw provides quality control oversight for the Roche Electrosys immunoassay platform as part of the ADNI-3 study. Dr. Weiner has served on the Scientific Advisory Boards for Alzheon, Inc., Accera, Merck, Nestle (Nolan), PCORI (PPRN), Eli Lilly, Delfino Logic Ltd. (for Merck), Dolby Ventures, Brain Health Registry, and ADNI. He served on the editorial boards for Alzheimer's & Dementia and MRI. He has provided consulting and/or acted as a speaker/lecturer to Synarc, Pfizer, Accera, Inc., Alzheimer's Drug Discovery Foundation (ADDF), Merck, BioClinica, Eli Lilly, Howard University, Guidepoint, Denali Therapeutics, Nestle/Nestec, GLG Research, Atheneum Partners, BIONEST Partners, American Academy of Neurology (AAN), and Society for Nuclear Medicine and Molecular Imaging (SNMMI). Other authors report no competing interests."

We deleted the conflict of interest section from the manuscript. We added the following statement to the financial disclosures and in the cover letter: “This does not alter our adherence to PLOS ONE policies on sharing data and materials.”

6. One of the noted authors is a group or consortium "the Alzheimer’s Disease Neuroimaging Initiative*". In addition to naming the author group, please list the individual authors and affiliations within this group in the acknowledgments section of your manuscript. Please also indicate clearly a lead author for this group along with a contact email address.

ADNI’s author list includes 15 pages of authors. We included a link to the PDF that includes the whole list of acknowledgements. We list Michael W. Weiner as the main contact for ADNI. We added to the manuscript:

Michael Weiner is the Principal Investigator for ADNI (michael.weiner@ucsf.edu). A full list of authors can be found at: https://adni.loni.usc.edu/wp-content/uploads/how_to_apply/ADNI_Acknowledgement_List.pdf.

7. Please amend your list of authors on the manuscript to ensure that each author is linked to an affiliation. Authors’ affiliations should reflect the institution where the work was done (if authors moved subsequently, you can also list the new affiliation stating “current affiliation:….” as necessary).

8. Please include your full ethics statement in the ‘Methods’ section of your manuscript file. In your statement, please include the full name of the IRB or ethics committee who approved or waived your study, as well as whether or not you obtained informed written or verbal consent. If consent was waived for your study, please include this information in your statement as well. 

There was no contact with patients, we used anonymized data (this was now added to the methods, as stated in section 2. We also indicated above and in the methods section of the manuscript that patients gave informed consent for the ADNI study.

Reviewers' comments:

Reviewer's Responses to Questions

Comments to the Author

Reviewer #1: This is well conducted study.

The author showed, Flortaucipir indices showed the strongest clinical association among the studied biomarkers and were predictive in the preclinical disease stages.

Among the flortaucipir indices, the machine-learning derived SPARE-Tau index was the most sensitive clinical progression biomarker.

The combination of different biomarker modalities better predicted cognitive performance.

Reviewer #2: The authors developed the novel machine learning derived index, SPARE-Tau, to assess tau PET accumulation in the earliest disease stage of Alzheimer’s disease (AD) continuum. The authors demonstrated that SPARE-Tau was the most sensitive clinical progression biomarker among all tested tau PET indices. This paper proposed the novel unbiased index of tau PET with flortaucipir as predictive biomarker in the preclinical disease stages of AD continuum to facilitate clinical research and clinical trials. However, there remains several concerns that should be addressed before considering publication.

Major points.

1. Previous studies reported that CSF p-Tau best predicted longitudinal cognitive decline in cognitively unimpaired subjects [X], while the authors showed no significant association of baseline CSF p-tau with longitudinal cognitive decline. Although predictive performance of tau PET could become better using the novel machine learning-derived index (SPARE-Tau) instead of other conventional indices, the reason why the predictive performance of CSF p-Tau discrepancies between the present and previous studies should be discussed.

X. Mattsson-Carlgren N, et al., Neurology. 2020;94(21):e2233-e2244.

There are several differences in between the sample and assays from the Mattsson-Carlgren manuscript and ours. The Mattsson-Carlgren manuscript included a longer follow-up (6 years), a difference CSF p-tau assay, and very different prevalence of abnormal tau biomarker values compared to our analysis. The Mattsson-Carlgren manuscripts had a 32% positive CSF p-tau versus a 5% positive inferior temporal SUVR (approximately 6 to 1 ratio). In our study, we had a 23.6% positive CSF p-tau versus a 12% positive inferior temporal SUVR (approximately 2 to 1 ratio). Therefore, we have a much smaller discrepancy between the biomarkers and a larger proportion of participants with abnormal SPARE-Tau values, which likely increases the statistical power to find differences. As discussed below, it is also possible that because tau PET changes happen later, close to onset of clinical symptoms, tau PET indices show a stronger association with shorter follow-ups. A later study from the same group that include BIOFINDER and ADNI results confirmed that “Replication of the analyses in ADNI revealed overall similar results, except that the stronger association of CSF with cognitive measures in CU individuals observed in BioFINDER-2 was not present in ADNI” (Ossenkoppele EMBO Mol Med (2021)13:e14398). In our analyses including all CU participants CSF p-tau predicted ADAS-Cog changes when using z-scored values and results almost reached significance for the cutoff analysis (p-value=0.059),

We now include the following comment in the discussion; “One study has described inconsistent findings of CSF p-tau better predicting cognition in CU participants than tau PET(44). These differences might be to differences in cohort composition, CSF assays, and length of follow-up. Further studies with longer longitudinal follow-up that include plasma, CSF and PET tau measures in CU participants in CU participants are needed.”

2. Recent study demonstrated that CSF p-tau generally becomes abnormal before tau PET and that cognitively unimpaired subjects with abnormal CSF p-Tau and positive PET scan have a risk of clinical progression, followed by cognitively unimpaired subjects with abnormal CSF p-Tau despite negative scan of Tau PET [Y]. In the present study, the median follow up was just 1.9 years, which is relatively short duration. Taken together, although we can conclude that elevated accumulation of Tau PET must predict future cognitive decline, we didn’t draw the robust conclusion whether the CSF p-Tau would predict future cognitive decline because of the short follow up period.

Y. Groot C, et al., Brain. 2022 Sep 9:awac329. doi: 10.1093/brain/awac329. Online ahead of print.

The manuscript by Groot et al groups together cognitively normal and MCI participants with an MCI % of 27% (n=48), 33% (n=30), and 72% (n=18) in the different A+ groups. This is a different group compared to our validation group which consists entirely of cognitively unremarkable (CU) A+ participants (n=135). The approach by Groot et al of including MCI makes it difficult to interpret associations in the preclinical stage. Regarding the follow-up the Groot el al study also had a 2-year follow-up, although due to the inclusion of MCI there might be a higher rate of progression especially for their A+P+T+ group which included 72% of MCI. We think that our evaluation including only CU A+ offers a better interpretation of the predementia prediction. In our sample there was a higher number of CU A+ participants with abnormal CSF p-tau values compared to abnormal SPARE-tau values, this might indicate that CSF p-tau might detect changes at an earlier stage. Therefore, with a short follow-up, CSF p-tau might underperform SPARE-tau. Unfortunately, the design of our current study and length of follow-up do not allow to offer a more definitive answer at this point.

We added two sentences to highlight this important point in the discussion:

Alternatively, it is possible that CSF p-tau identifies changes at an earlier preclinical stage than SPARE-Tau (23.6% abnormal CSF p-tau and 12% abnormal SPARE-Tau in the CU Aβ+ group). This could also explain why CSF p-tau underperforms SPARE-Tau in the case of a short follow-up. 

3. In the section of “MRI Acquisition and Processing” (page 7, lines 128-131), the authors explained that “The training data included only healthy controls with known-negative Aβ status and only dementia participants with known-positive Aβ status.”; however, then it is also written that “The model was trained with CU data only and applied to all participants included in this study”. What does it mean? Was SPARE-AD model retrained without data of dementia participants with known-positive Aβ status?

Thank you for bringing up this point. We omitted including the reference to SPARE-BA when we added this information from the supplementary material. We now modified this section which reads: “The SPARE-BA model was trained with CU data only and applied to all participants included in this study.” 

Minor points

4. Page 6, lines 115-117

Regarding the PET reading method, whether visual judgment was used or semi-quantitative values such as centiloid scale or SUVR were used, and if so, how the cutoff values were set should be briefly indicated.

This information was listed at the end of the “PET Acquisition and Processing” section, on page 8, lines 149-154. We added the ADNI webpage for further details and we added a reference on page 6 pointing towards CSF and PET sections for further details. We used the composite score using the white matter reference region developed by the Jagust lab, with their recommended cutoff of 0.78.

5. Page 8, lines 149-151

The “)” may be missing in the sentence.

We appreciate the attentive reading and bringing this to our attention. We corrected it as recommended.

6. Page 8, line 155

Only the title of the chapter (SPARE-AD Training) is given, the contents is not written.

We include the information that was previously listed in the supplementary material. 

7. Page 10, line 201

“CSF p=tau” shold be corrected to “CSF p-tau“.

We appreciate the attentive reading and bringing this to our attention. We corrected it as recommended.

---

## [Decision Letter · Decision Letter 1]

6 Oct 2022

SPARE-Tau: A Flortaucipir Machine-Learning Derived Early Predictor of Cognitive Decline

PONE-D-22-21443R1

Dear Dr. Habes,

We’re pleased to inform you that your manuscript has been judged scientifically suitable for publication and will be formally accepted for publication once it meets all outstanding technical requirements.

Kind regards,

Kensaku Kasuga

Academic Editor

PLOS ONE

Additional Editor Comments (optional):

Reviewers' comments:

Reviewer's Responses to Questions

**Comments to the Author**

1. If the authors have adequately addressed your comments raised in a previous round of review and you feel that this manuscript is now acceptable for publication, you may indicate that here to bypass the “Comments to the Author” section, enter your conflict of interest statement in the “Confidential to Editor” section, and submit your "Accept" recommendation.

Reviewer #2: All comments have been addressed

2. Is the manuscript technically sound, and do the data support the conclusions?

Reviewer #2: Yes

3. Has the statistical analysis been performed appropriately and rigorously? 

Reviewer #2: Yes

4. Have the authors made all data underlying the findings in their manuscript fully available?

Reviewer #2: Yes

5. Is the manuscript presented in an intelligible fashion and written in standard English?

Reviewer #2: Yes

6. Review Comments to the Author

Reviewer #2: The authors addressed all of my questions and concerns.

I believe the manuscript vastly improved.

I would have no further criticism.

7. PLOS authors have the option to publish the peer review history of their article (what does this mean?). If published, this will include your full peer review and any attached files.

Reviewer #2: **Yes: **Hitoshi SHIMADA

---

## [Editor Report · Acceptance letter]

24 Oct 2022

PONE-D-22-21443R1 

SPARE-Tau: A Flortaucipir Machine-Learning Derived Early Predictor of Cognitive Decline 

Dear Dr. Habes:

I'm pleased to inform you that your manuscript has been deemed suitable for publication in PLOS ONE. Congratulations! Your manuscript is now with our production department. 

Kind regards, 

on behalf of

Dr. Kensaku Kasuga 

Academic Editor

PLOS ONE